# Fabrication of Durable Superhydrophobic Surface for Versatile Oil/Water Separation Based on HDTMS Modified PPy/ZnO

**DOI:** 10.3390/nano12142510

**Published:** 2022-07-21

**Authors:** Shumin Fan, Sujie Jiang, Zhenjie Wang, Pengchao Liang, Wenxiu Fan, Kelei Zhuo, Guangri Xu

**Affiliations:** 1Postdoctoral Research Base, Henan Institute of Science and Technology, Xinxiang 453003, China; 2Postdoctoral Research Station, School of Chemistry and Chemical Engineering, Henan Normal University, Xinxiang 453007, China; 3School of Chemistry and Chemical Engineering, Henan Institute of Science and Technology, Xinxiang 453003, China; 15290088739@163.com (S.J.); wangzhenjie572@126.com (Z.W.); asd1346156@126.com (P.L.); fwxiu@hist.edu.cn (W.F.); 4School of Chemistry and Chemical Engineering, Henan Normal University, Xinxiang 453007, China

**Keywords:** superhydrophobic, oil/water separation, durability, reusability, fabric

## Abstract

Superhydrophobic materials have been widely applied in rapid removal and collection of oils from oil/water mixtures for increasing damage to environment and human beings caused by oil-contaminated wastewater and oil spills. Herein, superhydrophobic materials were fabricated by a novel polypyrrole (PPy)/ZnO coating followed by hexadecyltrimethoxysilane (HDTMS) modification for versatile oil/water separation with high environmental and excellent reusability. The prepared superhydrophobic surfaces exhibited water contact angle (WCA) greater than 150° and SA less than 5°. The superhydrophobic fabric could be applied for separation of heavy oil or light oil/water mixtures and emulsions with the separation efficiencies above 98%. The coated fabric also realized highly efficient separation with harsh environmental solutions, such as acid, alkali, salt, and hot water. The superhydrophobic fabric still remained, even after 80 cycles of separation and 12 months of storage in air, proving excellent durability. These novel superhydrophobic materials have indicated great development potentials for oil/water separation in practical applications.

## 1. Introduction

Increasing water pollution from offshore oil spills and industry activities has threatened the human beings and environment seriously [1,2]. The traditional treatment methods, including microbial degradation, oil containment fences, and in situ burning, have disadvantages of low efficiency, low selectivity, and cumbersome operation. Thus, finding an effective method to realize rapid collection of organic solvents and oils from water has become urgent. The superhydrophobic material, which shows different interfacial behaviors towards oil and water, has attracted significant attention in oil/water separation.

The superhydrophobic material is water-repellent, with a water contact angle (WCA) greater than 150° and a sliding angle (SA) smaller than 10°. The superhydrophobic surface could be realized by the following approaches: construction of a suitable roughness structure and application of low surface energy substances on rough surface. Various methods have been developed to prepare superhydrophobic surfaces, including sol–gel method [3], hydrothermal treatment [4], layer by layer assembly [5], electrochemical methods [6], chemical vapor deposition [7], plasma treatment [8], chemical etching [9], and self-assembly method [10]. The superhydrophobic materials were usually constructed on textile [11,12,13], fabrics [14], meshes [15], and so on.

In particular, the inorganic micro/nanoparticles have been used to create the roughness structure for superhydrobicity preparation, such as TiO_2_ [16], SiO_2_ [17], Fe_3_O_4_ [18], ZnO [19], and carbon nanotubes [20]. ZnO nanoparticles have significantly attracted interests due to their low cost, accessibility, and stability in application [21]. Barthwal et al. [22] introduced a superhydrophobic coating, which is composed of multi-walled carbon nanotubes/ZnO composite modified with polydimethylsiloxane. The coating showed excellent performance in self-cleaning, anti-fouling, and oil/water separation. Wei et al. [23] prepared superhydrophobic material for oils purification. The ZnO nanoflower modified SiC composite ceramic membranes were fabricated by chemical bath deposition method. Sun et al. [24] fabricated a superhydrophobic polyurethane sponge-based coating on a ZnO/epoxy resin solution followed by modification with stearic acid. Pal et al. [25] achieved superhydrophobic substrates based on Ag/ZnO/Ag hybrid structure. These methods for fabrication of superhydrophobic coating always require a time-consuming and tedious process. Most methods faced common drawbacks, such as air pollution, poor bonding strength between substrate and coating, specialized reagent, and harsh condition requirements [26,27,28,29]. In addition, most of the superhydrophobic materials could be damaged under corrosive solutions and harsh conditions, which has greatly limited their application in practical separation of real oil with harsh aqueous solutions [30]. Therefore, developing a novel superhydrophobic material with simple process, high environmental ability, and durability is of great importance for its practical applications in oil/water separation from chemical leakages and oil spills.

For realizing the low surface energy on rough surface, hexadecyltrimethoxysilane (HDTMS) is highly used for preparation of superhydrophobic material in oil/water separation [31,32]. Chen et al. [33] coated the flax fiber with ZnO-HDTMS using a plasma-grafted poly (acrylic acid) as the binding agent for efficient oil/water separation. Zhang et al. [34] bonded ZnO nanoparticles to the surface of polyurethane/polysulfone with hydrophobically modification of HDTMS to prepare a new waterproof and moisture permeable membrane.

In this paper, we report a facile, inexpensive, but versatile method for preparation of superhydrophobic surfaces with high durability. The samples were integrated with PPy and ZnO nanoparticles by in situ polymerization and deposition method. PPy/ZnO was loaded onto different substrates, which constituted an appropriate hierarchical structure. After further modification with HDTMS, the samples became superhydrophobic and superoleophilic, which could be used for oil/water mixtures separation. This method had the following apparent advantages: (1) There were no restrictions on the shape or type of the substrates. Fabric, sponge, copper mesh, forestry, and agricultural residues could all be applied without destroying the intrinsic appearance; (2) This method was economical and efficient because it was conducted at low temperature without special materials or equipment; (3) The superhydrophobic coating showed low-term durability and mechanical stability in oil/water separation; (4) This method could prepare superhydrophobic materials on a large scale. Apparently, a simple and feasible strategy has been put forward to fabricate superhydrophobic surfaces, which provides a feasible solution for oil/water separation.

## 2. Materials and Methods

### 2.1. Materials

Corn stalk, sawdust, cotton, fabric, melamine sponge, and copper mesh were purchased from a local market. The cotton sample is the raw fiber. The fabric used is the cotton gauze, which has wrap and weft yarns of 32 counts of pure cotton with a number of 50 × 44 per square inch. Zinc chloride (ZnCl_2_, 99%), pyrrole (C_4_H_4_NH, 99%), hexadecyltrimethoxysilane (HDTMS, CH_3_(CH_2_)_15_Si(OCH_3_)_3_, ≥85%), ammonium hydroxide (NH_3_∙H_2_O, 28%), absolute ethyl alcohol (C_2_H_5_OH), acetic acid (CH_3_COOH, 99.5%), ethyl acetate (C_4_H_8_O_2_, 99%), liquid paraffin (99%), benzene (C_6_H_6_, ≥99.5%), petroleum ether, tetrachloromethane (CCl_4_, 98%) methylbenzene (C_7_H_8_, 99%), chloroform (CHCl_3_, 99%), and dichloromethane (CH_2_Cl_2_, 99%) were acquired from Shanghai Macklin Biochemical Co., Ltd. (Shanghai, China). Sodium hydroxide (NaOH, ≥96.0%), hydrochloric acid (HCl, 36.5%∼38%), and ferric trichloride (FeCl_3_, 99%) were purchased from Tianjin Kermel Chemical Reagent Co., Ltd. (Tianjin, China). The engine oil, diesel, and corn oil were obtained locally.

### 2.2. Preparation of Superhydrophobic Coating

The superhydrophobic coating was fabricated on various substrates with different raw materials, including corn stalk, sawdust, cotton, fabric, sponge, and copper mesh. Taking the preparation of superhydrophobic fabric as an example, the fabric specimens (40 mm × 40 mm) were washed in ethanol aqueous solution (2%) and immersed in 0.4 mol/L pyrrole aqueous solution under stirring for 2 h. An equal volume of 0.15 mol/L FeCl_3_ was added subsequently to be placed at 5 a temperature of 5 °C for 80 min. After immersion in ethanol aqueous solution and drying, the PPy coated fabric was acquired.

The PPy coated fabric was add into the mixture of 0.1 mol/L ZnCl_2_ and 10 mol/L of ammonium hydroxide, which was sealed at 75 °C for 4 h. The sample was dried to obtain PPy/ZnO coated fabric. Additionally, the PPy/ZnO coated fabric was immersed in HDTMS–ethanol solution (the mixture of 2 mL HDTMS, 100 mL anhydrous ethanol, 0.25 mL deionized water, and 0.05 mL acetic acid) for 4 h. Finally, the PPy/ZnO/HDTMS coated fabric was obtained after further drying.

### 2.3. Oil/Water Separation

The immiscible oil/water mixtures were acquired by mixing the oils and water directly. The water-in-tetrachloromethane emulsion was obtained by mixing 1 mL water, 114 mL tetrachloromethane, and 0.5 g Span 80 under stirring for 3 h. In order to separate oil/water mixtures, the superhydrophobic fabric was fixed between two identical tubes with a clamping device. The oil filtrate fraction *η* (%) was calculated by Equation (1):(1)$$\eta =M_{2}⁄M_{1}\;$$
where *M*_1_ is oil mass in original oil/water mixture and *M*_2_ is the collected oil mass after separation.

The oil flux *F_oil_* was calculated by Equation (2):(2)$$F_{oil}=V/St$$
where *V* represents the oil volume in oil/water mixture, *S* represents the cross-sectional area of the superhydrophobic fabric exposed to oil/water mixtures, and *t* represents the time needed for complete separation.

The oil absorption capacity of the superhydrophobic fabric was analyzed by immersed the prepared fabric in different organic solvents and oils for saturation. The oil absorption capacity *Q* was calculated by Equation (3):(3)$$Q=\left(m_{2}-m_{1}\right)/m_{1}$$
where *m*_1_ and *m*_2_ denote the mass of prepared fabric before and after absorption, respectively.

### 2.4. Characterization

The apparent WCA and SA on samples were measured by an TST-200H optical contact angle meter (Shenzhen testing equipment CO., LTD, Shenzhen, China). The WCA was measured using a 5 μL of deionized water droplet, which was determined by averaging 5 values from different positions of each sample. The surface morphologies of samples were observed with a Quanta 200 scanning electron microscope (SEM, FEI, Hillsboro, USA). The chemical compositions were characterized using a Magna-IR 560 fourier transform infrared spectrum (FTIR, Thermo Nicolet, Madison, USA). The FTIR data were collected at room temperature in the 4000−400 cm^−1^ range (30 scans at a resolution of 4 cm^−1^).

## 3. Results and Discussion

### 3.1. Surface Morphology and Wettability

Micro/nanostructures’ creation and modification of low surface energy material are of importance to realize superhydrophobicity. The surface morphologies of raw and coated samples were investigated. Figure 1a,d,g,j,m,p showed the typical images of original corn stalk, sawdust, cotton, fabric, sponge, and copper mesh. After the coating treatment, the surface of all samples became hierarchical and rough. From high magnifications, ZnO nucleated on the surface and grew in addition to precipitation or in lieu of precipitation. The irregular distribution of particles covered the surfaces of samples uniformly, which created micro/nanostructures required by superhydrophobicity.

The PPy/ZnO/HDTMS coating was analyzed with the Cassie–Baxter equation [35]:(4)$$cos\theta_{c}=f_{s}\left(cos\theta +1\right)-1$$

In this equation, *θ_c_* and *θ* are the apparent WCA of a water droplet on rough and smooth solid surface, respectively. *f_s_* is the apparent area fraction of the solid surface in contact with water. Thus, 1−*f_s_* is apparent area of trapped air contacting with water. Table 1 showed the calculated data with the Cassie–Baxter equation according to the wettability of PPy/ZnO/HDTMS coated samples. The WCA of original corn stalk, sawdust, cotton, fabric, sponge, and copper mesh were 0°, 0°, 0°, 0°, 90.9°, and 85.2°, respectively. The WCA of PPy/ZnO/HDTMS coated samples were 152.7°, 153.9°, 153.3°,157.6°, 163.1°, and 162.1°, respectively. Specifically, the fraction of air (1−*f_s_*) in contact with water of all coated samples was above 0.88, illustrating that the air occupied more than 88% of the contact area when contacting with water droplets on the coated rough surface. Thus, the air preserved in hierarchical texture mainly supported the sphere-shaped water droplet.

The photographs of water droplets on original and PPy/ZnO/HDTMS coated samples were shown in Figure 2. All coated samples presented dark surface because of the first step of PPy modification. From Figure 2, the water droplets could keep a spherical shape on surface of coated samples, which was consistent with the calculated data in Table 1. Therefore, it is proved that the coated samples became superhydrophobic. The modification of ZnO nanoparticles and HDTMS are of importance to the endowment of superhydrophobicity on different substrates.

### 3.2. Formation Mechanism

As shown in Figure 3, the FTIR spectra were tested in order to ensure the chemical compositions of the prepared coating. For raw fabric, the strong absorption peak at 3424 cm^−1^ was attributed to -OH stretching vibration. The -OH stretching vibration was greatly reduced after coating treatment due to the modification of HDTMS. The FTIR spectrum for PPy [36,37,38] and HDTMS [39] is shown in Table 2. The characteristic peaks at 1685, 1558, and 1315 cm^−1^ in coated fabric were assigned to the C = N, C = C, C–N in PPy. Two absorption peaks at 2851 and 2921 cm^−1^ stemmed from C–H stretching vibration of –CH_2_– and –CH_3_. The adsorption at 2918 cm^−1^ (C–H stretching vibration) in original fabric split into two peaks (2921 and 2851 cm^−1^) after coating, which was attributed to HDTMS modification. Furthermore, the stretching vibration of Si–C was observed at 781 cm^−1^, demonstrating that the HDTMS was successfully grafted onto the PPy/ZnO coated fabric.

The XRD diffraction patterns of raw and coated fabric were shown in Figure 4. The raw fabric had typical diffraction peaks of cellulose at 2θ = 14.9°, 16.5°, and 22.9°, which also existed in coated fabric. The diffraction peaks of the coated fabric at 32.45°, 34.76°, 36.82°, and 47.65° corresponded to the (100), (002), (101), and (102) lattice planes of ZnO nanoparticles, proving that the ZnO nanoparticles were successfully loaded on fabric surface. ZnO nanoparticles played a significant role in constituting the hierarchical structure for realizing the superhydrophobic properties.

The reaction steps of the PPy/ZnO/HDTMS coating are shown in Figure 5a. PPy combined with the fabric in the first by in situ polymerization reaction. ZnO nanoparticles were deposited on PPy surface by precipitation method. From Figure 5b, the hydrolysis of HDTMS generated –OH groups which replaced the –O–CH_3_ groups. The –OH groups on ZnO nanoparticles or formed by hydrolysis of HDTMS were both reactive, which resulted in the condensation reaction between them. Furthermore, the –OH groups of hydrolyzed HDTMS reacted with each other, forming the Si–O–Si bond. Analogously, HDTMS could be chemically bonded with PPy through the condensation reaction between the –OH groups in hydrolyzed HDTMS and N–H bond in PPy to further reduce the surface energy.

### 3.3. Oil/Water Separation

The PPy/ZnO/HDTMS coated fabric with superhydrophobic and superolephilic properties was used for oil/water mixtures separation. The separation process driven by gravity is shown in Figure 6. Heavy oil like tetrachloromethane and light oil like petroleum ether were used for separation. The oil/water mixtures were poured from the top crossing the coated fabric which was fixed between two tubes. For tetrachloromethane/water separation, the separation device was placed with a vertical type (Figure 6a and Appendix A). The water flowed into the tube first but could not permeate through the coated fabric for its superhydrophobicity. The tetrachloromethane, arriving late, passed through the water phase and penetrated into the coated fabric because of the higher density of tetrachloromethane and the superoleophilicity of the fabric, enabling the heavy oil/water mixture to be completely and successfully separated. The oil filtrate fraction and flux of the coated fabric for tetrachloromethane/water mixture were calculated to be 99.3% and 782.4 L m^−2^ h^−1^, respectively. For petroleum ether/water separation, the separation device was placed with an inclined type. As shown in Figure 6b and Appendix A, the petroleum ether encountered and passed through the coated fabric first, while the water could not submerge the coated fabric, which resulted in a complete separation of light oil/water mixture. With further evaluation and calculation, the separation and flux of the coated fabric for petroleum ether/water mixture were 98.4% and 426.7 L m^−2^ h^−1^, respectively. Other mixtures could also be separated using the coated fabric efficiently, for example, dichloromethane/water mixture (99.4%, 855.6 L m^−2^ h^−1^), the diesel/water mixture (99.3%, 432.9 L m^−2^ h^−1^), and ethyl acetate/water mixture (98.6%, 598.4 L m^−2^ h^−1^). Moreover, the coated fabric also had an outstanding separation effect for water-in-oil emulsion, and the filtration procedure and effects were also investigated. As shown in Figure 6c and Appendix A, the water-in-tetrachloromethane emulsion was slowly filtered owing to gravity. The coated fabric was wet with the oil phase which fell into the measuring beaker. Finally, the foggy cloudy emulsion became transparent and clarified. The micron water droplets in emulsion before filtration was evaluated to be 2–5 μm, which completely disappeared after separation (Figure 6d). The oil filtrate fraction and flux of the coated fabric for the water-in-tetrachloromethane emulsion were found to be 99.1% and 736.9 L m^−2^ h^−1^. Except for the water-in-tetrachloromethane emulsion, other emulsions could also be separated efficiently, for example, the water-in-chloroform emulsions (99.2%, 853.6 L m^−2^ h^−1^), methylbenzene-in-water emulsion (99.1%, 822.4 L m^−2^ h^−1^). In conclusion, the PPy/ZnO/HDTMS coated fabric exhibited an outstanding separating performance towards different water/oil mixtures, which could satisfy different needs in practical usage, showing its great potential.

The oil absorption capacity of the coated fabric was analyzed, which is shown in Table 3. The oil absorption capacity towards different oils differed for the physical and chemical properties of oils. The original fabric had absorption capacity for ethyl acetate of 4.83 g/g, while the oil absorption capacity of the coated fabric was 9.61 g/g, which was almost two times that of the original fabric. The as-prepared fabric could absorb oils up to many times its own mass. The fabric after coating treatment could absorb more oil. The roughness provided by PPy/ZnO and the low surface obtained by HDTMS obviously enhanced the absorption capacity of fabric for oils.

### 3.4. Durability Evaluation

Corrosive solutions, such as salt, acidic, basic, or hot solutions, are the existing challenges for practical application of superhydrophobic material in oil/water separation. The wettability of the coated fabric toward corrosive solutions was examined and shown in Figure 7a. The WCA of hot water (100 °C), 1M of HCl solution, 1M of NaOH solution, and 1M of NaCl solution were all above 150°, showing excellent superhydrophobic property of the coated fabric. The separation capacity of the coated fabric for oil/water separation under harsh conditions was subsequently evaluated. The mixtures of tetrachloromethane and various corrosive solutions were used for evaluation. The oil phase passed through the coated fabric quickly, leaving the corrosive solutions above the coated fabric. As shown in Figure 7b, the separation of these tetrachloromethane/corrosive solution mixtures was all completely achieved with the oil filtrate fraction above 98%. Thus, the coated fabric could realize highly efficient separation even in harsh environments, indicating its excellent capability in oil/water separation.

The superhydrophobic fabric also showed an outstanding durability. As shown in Figure 8a, the reusability of the coated fabric was evaluated by duplicating the oil/water separation for 80 times. The oil filtrate fraction of tetrachloromethane/water mixture after 80 separation cycles remained above 98%. The surface morphology of the coated fabric after being reused for 80 cycles is depicted in Figure 8b, and no changes were observed with this fabric, proving excellent reusability of the superhydrophobic fabric for oil/water separation. Figure 8d showed the abrasion resistance of the coated fabric. The superhydrophobic fabric was placed on a sandpaper with 1000 meshes, which was pulled forth and back under a weight of 100 g for a distance of 10 cm. The WCA of the coated fabric after 80 abrasion cycles was still above 150°, demonstrating outstanding mechanical robustness. Figure 8c showed the surface morphology of the coated fabric after 80 abrasion cycles. The surface was still rough and hierarchical, which was critical for keeping superhydrophobic properties. The firm modification of the superhydrophobic coating was responsible for the respectable durability of the coating. The storage stability test of the coated fabric in air was conducted. As depicted in Figure 8e, the coated fabric showed a stable superhydrophobicity with the WCA higher than 155° after being exposed in air for 12 months. All these characterizations demonstrated durable stability of the coated fabric for oil/water separation.

### 3.5. Comparison of the As-Prepared Fabric with Other Reported Superhydrophobic Fabrics

A comparison of the as-prepared fabric with other reported superhydrophobic fabrics in preparation and separation was made to demonstrate the significance of this work (Table 4). The other methods for fabrication of superhydrophobic coating always require a time-consuming and tedious process. The method proposed in this work showed obvious advantages of low cost and easy to operate. Additionally, both heavy and light oils could be separated from oil/water mixtures. As for separation under harsh environments, the as-prepared fabric could maintain its superhydrophobicity and oil filtrate fraction under specific harsh conditions, demonstrating outstanding performance for highly efficient separation in harsh environments. Generally, the as-prepared fabric showed superior property compared with other superhydrophobic fabrics.

## 4. Conclusions

In summary, a novel PPy/ZnO/HDTMS coated superhydrophobic surface has been presented which was capable of oil/water separation even in harsh environments. The PPy/ZnO coating was prepared by in situ polymerization and deposition method, which was an effective method for constructing nanometer-scale grains on different substrates. The as-prepared materials exhibited superhydrophobic and superoleophilic properties, which were applied for immiscible oil/water mixtures and emulsions with separation efficiencies above 98% with a high purity of collected oils. Moreover, the superhydrophobic material also exhibited outstanding reusability and stability through 80 repeated separations of tetrachloromethane/water mixture and 12 months of exposure to air. The reported superhydrophobic surface has displayed strong potential for versatile oil/water separation needed for practical applications. The superhydrophobic material prepared in this work displayed outstanding performance for removing/collecting oil contaminants. This strategy for achieving superhydrophobic material is simple, feasible, and highly efficient, offering a wide range of potential applications in reclaiming the oil/water mixtures from chemical leakages and oil spills.

## Figures and Tables

**Figure 1 nanomaterials-12-02510-f001:**
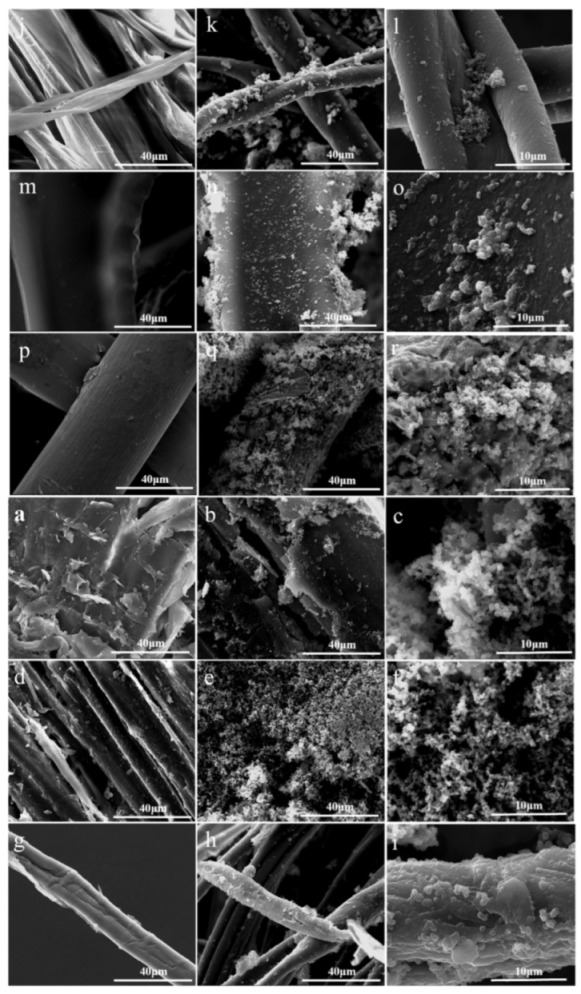
SEM images of raw and PPy/ZnO/HDTMS coated samples. (**a**,**d**,**g**,**j**,**m**,**p**): corn stalk, sawdust, cotton, fabric, sponge, copper mesh; (**b**,**e**,**h**,**k**,**n**,**q**): the coated samples; (**c**,**f**,**i**,**l**,**o**,**r**): higher magnification images of the coated samples.

**Figure 2 nanomaterials-12-02510-f002:**
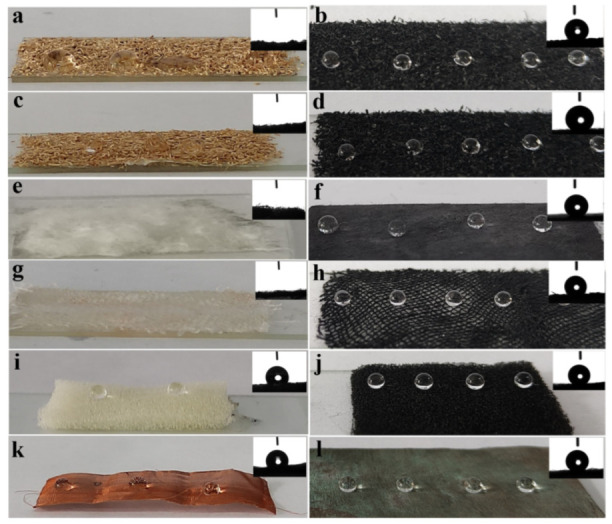
Photographs of original (**left**) and PPy/ZnO/HDTMS coated samples (**right**). (**a**,**b**): corn stalk; (**c**,**d**): sawdust; (**e**,**f**): cotton; (**g**,**h**): fabric; (**i**,**j**): sponge; (**k**,**l**): copper mesh.

**Figure 3 nanomaterials-12-02510-f003:**
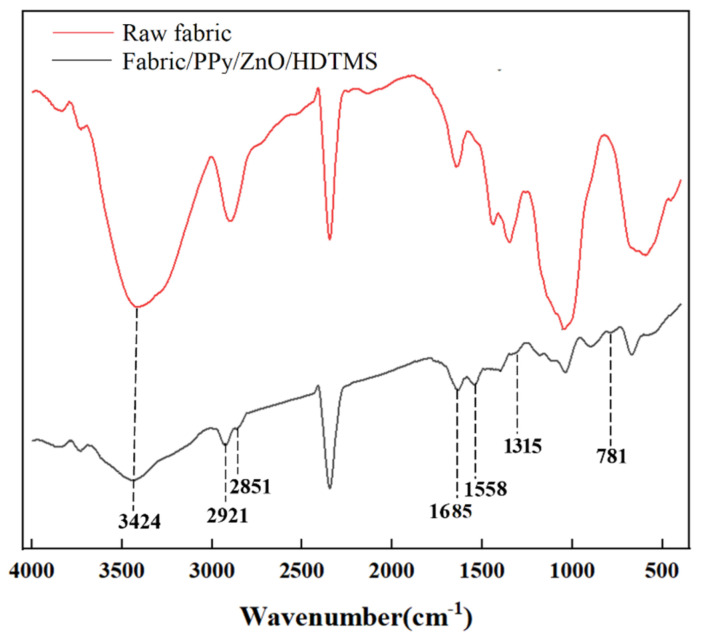
FTIR spectrum.

**Figure 4 nanomaterials-12-02510-f004:**
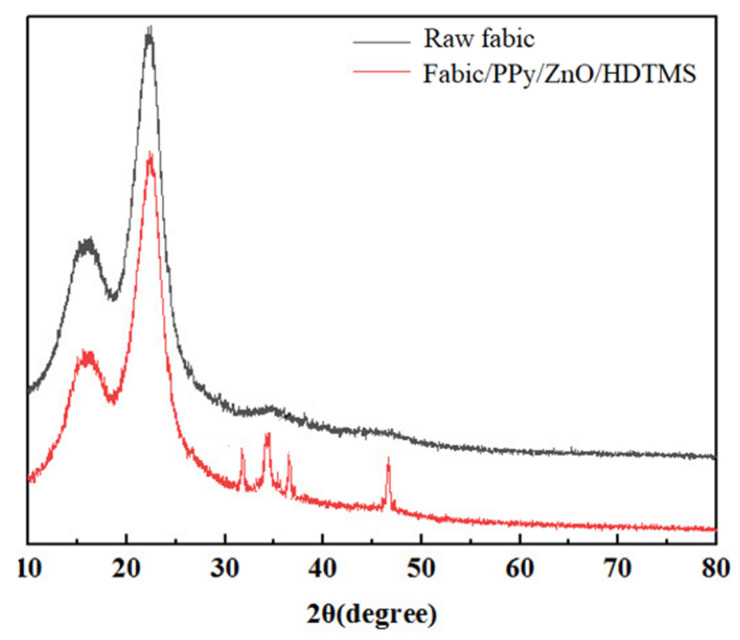
XRD patterns.

**Figure 5 nanomaterials-12-02510-f005:**
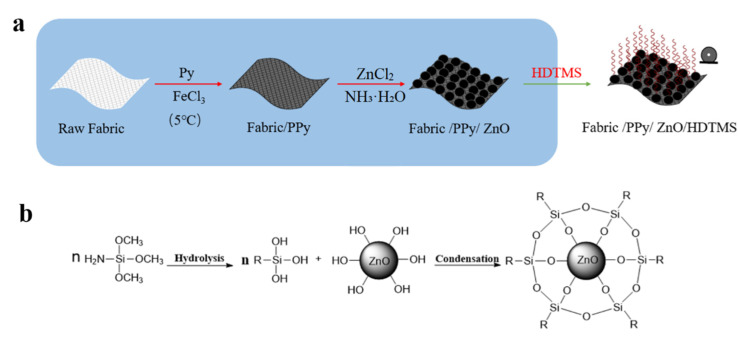
(**a**) Reaction steps of the PPy/ZnO/HDTMS coating; (**b**) A reaction mechanism diagram for HDTMS grafting on ZnO nanoparticles.

**Figure 6 nanomaterials-12-02510-f006:**
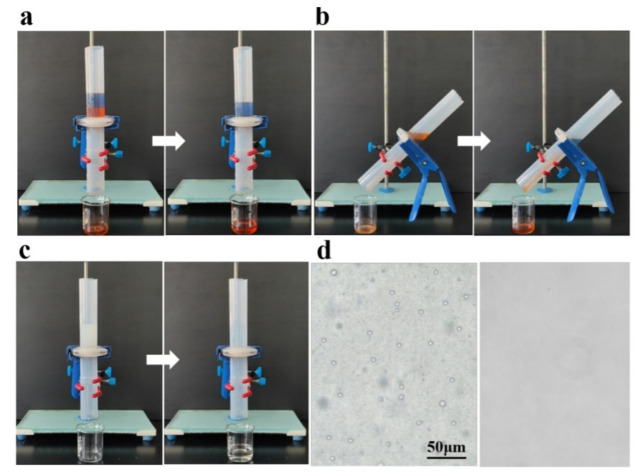
Oil/water separation. (**a**), tetrachloromethane/water mixture; (**b**), petroleum ether/water mixture; (**c**), water-in-tetrachloromethane emulsion; (**d**), the emulsion under optical microscope before (**left**) and after (**right**) separation.

**Figure 7 nanomaterials-12-02510-f007:**
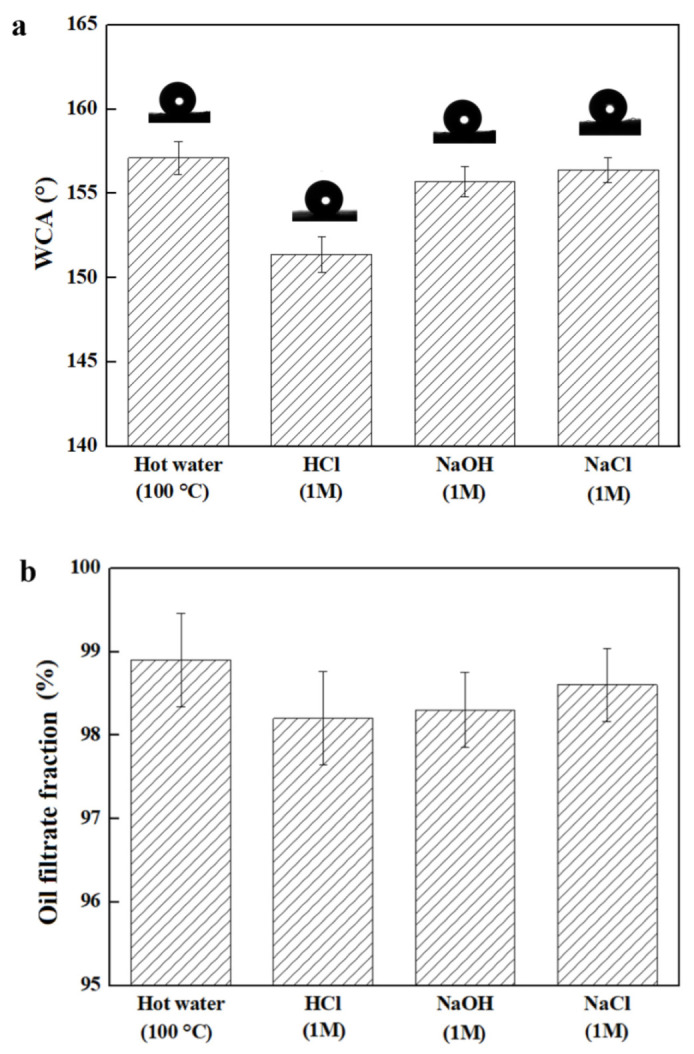
The resistance to corrosive solutions of the coated fabric. (**a**) The WCA of 5 μL of hot water (100 °C), HCl solution (1M), NaOH solution (1M), and NaCl solution (1M) on coated fabric; (**b**) The oil filtrate fraction of tetrachloromethane/corrosive solution mixtures.

**Figure 8 nanomaterials-12-02510-f008:**
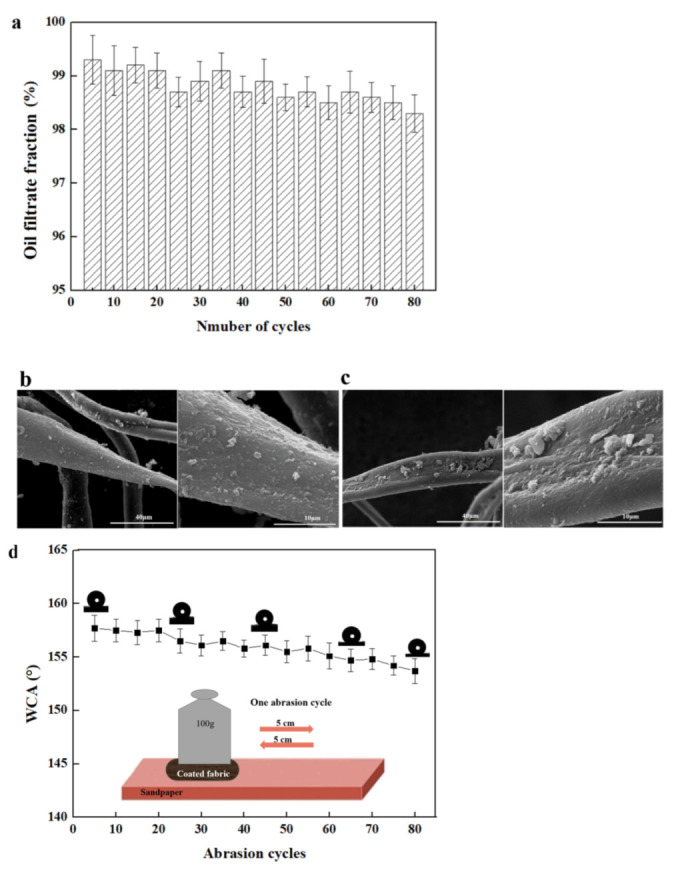
Durability of the coated fabric. (**a**) The oil filtrate fraction with number of cycles; (**b**) The SEM images of the coated fabric after being reused for 80 cycles; (**c**) The SEM images of the coated fabric after 80 abrasion cycles; (**d**) The WCA of coated fabric with abrasion cycles; (**e**) The WCA of coated fabric with the storage time in air.

**Table 1 nanomaterials-12-02510-t001:** Calculated data based on the Cassie–Baxter equation.

Substrate	WCA	SA	Air Fraction (1−*f_s_*)
corn stalk/PPy/ZnO/HDTMS	152.7 ± 1.6	2.0 ± 0.6	0.88
sawdust/PPy/ZnO/HDTMS	153.9 ± 2.3	3.5 ± 1.5	0.89
cotton/PPy/ZnO/HDTMS	153.3 ± 1.1	3.0 ± 1.7	0.88
fabric/PPy/ZnO/HDTMS	157.6 ± 4.5	3.8 ± 1.8	0.92
sponge/PPy/ZnO/HDTMS	163.1 ± 4.6	2.3 ± 0.8	0.95
copper mesh/PPy/ZnO/HDTMS	162.1 ± 3.3	3.9 ± 2.3	0.95

**Table 2 nanomaterials-12-02510-t002:** The FTIR spectrum for PPy and HDTMS.

Molecule	Wavelength (cm^−1^)	Attribution of the Characteristic Peaks
PPy	3522	N–H bonding in the molecule
	1685	C=N bonds in the molecule
	1558 1437	C=C bonds in the molecule
	1315	C-N bonds in the molecule
	811 920	Bonding of C–H in the molecule
HDTMS	2921	Asymmetric stretching vibration of –CH_2_
	2851	Symmetric stretching vibration of –CH_2_
	1000–1100	Bending vibration of Si–O–C
	781	Stretching vibration of Si–C

**Table 3 nanomaterials-12-02510-t003:** The oil absorption capacity of the coated fabric (g/g).

Oils	Ethyl Acetate	Liquid Paraffin	Benzene	Engine Oil	Corn Oil
Raw fabric	4.83	6.75	3.92	5.98	5.83
Fabric/PPy/ZnO/HDTMS	9.61	11.38	6.25	10.33	9.65

**Table 4 nanomaterials-12-02510-t004:** Comparison of preparation and separation with some reported works.

Materials	Fabrication Method	Efficiency (%)	Harsh Environment	References
Tetraethoxysilane/1,1,1,3,3,3-hexamethyl disilazane@PET textiles	sol–gel	none	none	[11]
Phytic acid-metal/ polydimethylsiloxane@ fabric	dip-coating	above 95.0	immersion in ethanol, n-hexane, xylene, and acetone	[12]
Polydopamine/ZIF-8@cellulose membrane	self-assembly	Above 99	immersion in water	[40]
Polydimethylsiloxane/ZIF-90/fluoroalkyl silane@linen fabric	dip-coating	99.5	immersion in 0.1 M H_2_SO_4_, 0.1 M NaOH, and salt solution	[41]
Polydopamine/Fe/hexadecyltrimethoxysilane@PET fabric	impregnation	above 95	immersion in organic reagents, seawater, HCl, and NaOH solution with pH of 1, 3, 5, 7, 9, 11, 13	[42]
Fabric/PPy/ZnO/HDTMS	impregnation	above 98.4	Immersion in hot water (100 °C), 1M of HCl solution, 1M of NaOH solution, and 1M of NaCl solution	this work

## Data Availability

The data presented in this study are available in this article.

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
