# Peer review of "Fabrication of Durable Superhydrophobic Surface for Versatile Oil/Water Separation Based on HDTMS Modified PPy/ZnO"

_nanomaterials, 2022, doi:10.3390/nano12142510_

Round 1

Reviewer 1 Report

In the study reported, a variety of substrates including corn stalk, sawdust, cotton, fabric, sponge, and copper mesh have been modified in a multistep chemical process to make them superhydrophobic and superoleophilic.  The process begins with polymerization of pyrrole, followed by formation of ZnO on the surface, and finally treatment with hexadecyltrimethoxysilane (HTDMS) to silanate the zinc oxide.  The modified fabric was examined for bulk phase separation of carbon tetrachloride/water and of petroleum ether/water as well as separation of an emulsion of water in carbon tetrachloride..   

Much work has been done on preparation of superhydrophobic materials for oil/water separation.  The technique described appears to be versatile in the types of materials that can be modified. Their method looks promising in that regard. Different length scales as required for superhydrophobicity are obtained.  

The demonstrated separation of the water in oil emulsion with use of the modified fabric as a filtration membrane is the most interesting and potentially useful result. Unfortunately, one cannot really tell how effective the separation is with the data as presented.  Without some measure of water passing through the membrane, the separation efficiency given by equation 1 is useless. It fails to indicate the amount of water removed.  With this definition, you can get a very high value for the separation efficiency, even if NO water is removed!  For a paper focused on materials for oil/separation, this must be addressed in a revision with a non-ambiguous measure of separation efficiency.

The bulk phase separations by filtration are not practical since a settling tank can be used or, alternatively, a separatory funnel at lab scale.    High absorption values are obtained for the modified fabric.  These are interesting findings, but should be compared to results obtained by others with similar materials.

Since much of the manuscript deals with a modified textile, the literature review should have more references in the area.  Here are some suggestions:

1. Xue CH, Ji PT, Zhang P, Li YR, Jia ST. Fabrication of superhydrophobic and superoleophilic textiles for oil–water separation. Applied Surface Science. 2013 Nov 1;284:464-71.

2. Zhou C, Chen Z, Yang H, Hou K, Zeng X, Zheng Y, Cheng J. Nature-inspired strategy toward superhydrophobic fabrics for versatile oil/water separation. ACS applied materials & interfaces. 2017 Mar 15;9(10):9184-94.

3. Yu H, Wu M, Duan G, Gong X. One-step fabrication of eco-friendly superhydrophobic fabrics for high-efficiency oil/water separation and oil spill cleanup. Nanoscale. 2022;14(4):1296-309.

Here is a reference to a recent article in Nanomaterials.

Song J, Liu N, Li J, Cao Y, Cao H. Facile fabrication of highly hydrophobic onion-like candle soot-coated mesh for durable oil/water separation. Nanomaterials. 2022 Feb 24;12(5):761.

There are a number of problems with Figure 2.  Values for the WCA in the text(line 145) do not match with what is shown in Figure 2.  For example, the contact angle for the unmodified fabric is given as 0 degrees, but this is clearly not the case for Fig. 2g.   Three or four of the six starting materials in Figure 2 are clearly hydrophobic while the text indicate four of them are hydrophilic.  Figure 2e does not look like cotton. 

A native speaker of English should edit the manuscript.  There are issues with capitalization, punctuation, spacing, misspellings and awkward wording.

Other points:

Define abbreviations given in the abstract.

In the Materials section, be more specific in your characterization of the substances to be modified.  Is the “cotton” sample raw fibers or a woven textile?   Specify the type of "fabric" used(e.g. polyester or silk) and give information on the fiber diameter and fiber density.   What is the chemical nature of the sponge?

Acetic acid does not appear in the materials list.  

XRD is mentioned, but no findings are reported.

Line 38:  “…and application of low surface energy substances…”

Line 176:  Figure 4a presents reaction steps, not reaction mechanisms.

Line 178:  The high resolution SEM images might suggest that ZnO nucleates on the surface and grows in addition to precipitation or in lieu of precipitation.

Author Response

Responses to Reviewer 1:

In the study reported, a variety of substrates including corn stalk, sawdust, cotton, fabric, sponge, and copper mesh have been modified in a multistep chemical process to make them superhydrophobic and superoleophilic.  The process begins with polymerization of pyrrole, followed by formation of ZnO on the surface, and finally treatment with hexadecyltrimethoxysilane (HTDMS) to silanate the zinc oxide.  The modified fabric was examined for bulk phase separation of carbon tetrachloride/water and of petroleum ether/water as well as separation of an emulsion of water in carbon tetrachloride..   

Much work has been done on preparation of superhydrophobic materials for oil/water separation.  The technique described appears to be versatile in the types of materials that can be modified. Their method looks promising in that regard. Different length scales as required for superhydrophobicity are obtained.  

Point 1: The demonstrated separation of the water in oil emulsion with use of the modified fabric as a filtration membrane is the most interesting and potentially useful result. Unfortunately, one cannot really tell how effective the separation is with the data as presented.  Without some measure of water passing through the membrane, the separation efficiency given by equation 1 is useless. It fails to indicate the amount of water removed.  With this definition, you can get a very high value for the separation efficiency, even if NO water is removed!  For a paper focused on materials for oil/separation, this must be addressed in a revision with a non-ambiguous measure of separation efficiency.

Response 1: During separation of water-in-oil emulsion, water could not pass through the coated fabric for its superhydrophobic property. Fig. 5d shows the emulsion under optical microscope before and after filtration. The optical microscope proved that the water in emulsion could not pass through the coated fabric. The micron water droplets in emulsion before filtration was evaluated to be 2-5 μm, which completely disappeared after separation. The oil phase fell into the measuring beaker, while water was prevented, achieving the separation of emulsion. Thus, equation 1 was used to evaluate the separation efficiency.

Point 2: The bulk phase separations by filtration are not practical since a settling tank can be used or, alternatively, a separatory funnel at lab scale. High absorption values are obtained for the modified fabric.  These are interesting findings, but should be compared to results obtained by others with similar materials.

Response 2: The absorption capacity of fabric was rarely studied. Thus, the as-prepared fabric was compared with other reported superhydrophobic fabric in preparation and separation. This part was added in manuscript in line 319-331.                

A comparison of the as-prepared fabric with other reported superhydrophobic fabric in preparation and separation was made to demonstrate the significance of this work (Table 4). The other methods for fabrication of superhydrophobic coating always require a time-consuming and tedious process. The method proposed in this work showed obvious advantages of low cost and easy to operate. Additionally, both heavy and light oils could be separated from oil/water mixtures. As for separation under harsh environments, the as-prepared fabric could maintain its superhydrophobicity and separation efficiency under specific harsh conditions, demonstrating outstanding performance for highly efficient separation in harsh environments. Generally, the as-prepared fabric showed superior property compared with other superhydrophobic fabrics.

Table 4. Comparison of preparation and separation with some reported works

Materials

Fabrication method

Efficiency (%)

Harsh environment

References

Tetraethoxysilane/1,1,1,3,3,3-hexamethyl disilazane@PET textiles

sol-gel

none

none

11

Phytic acid-metal/

polydimethylsiloxane@ fabric

dip-coating

above 95.0

immersion in ethanol, n-hexane, xylene and acetone

12

Polydopamine/ZIF-8@cellulose membrane

self-assembly

Above 99

immersion in water

41

Polydimethylsiloxane /ZIF-90/ fluoroalkyl silane@linen fabric

dip-coating

99.5

immersion in 0.1 M H2SO4, 0.1 M NaOH and salt solution

42

Polydopamine/Fe/ hexadecyltrimethoxysilane@PET fabric

impregnation

above 95

immersion in organic reagents, seawater, HCl and NaOH solution with pH of 1, 3, 5, 7, 9, 11, 13

43

Fabric/PPy/ZnO/HDTMS

impregnation

above 98.4

immersion in hot water (100 °C), 1M of HCl solution, 1M of NaOH solution and 1M of NaCl solution

this work

Point 3: Since much of the manuscript deals with a modified textile, the literature review should have more references in the area.  Here are some suggestions:

  1. Xue CH, Ji PT, Zhang P, Li YR, Jia ST. Fabrication of superhydrophobic and superoleophilic textiles for oil–water separation. Applied Surface Science. 2013 Nov 1;284:464-71.
  2. Zhou C, Chen Z, Yang H, Hou K, Zeng X, Zheng Y, Cheng J. Nature-inspired strategy toward superhydrophobic fabrics for versatile oil/water separation. ACS applied materials & interfaces. 2017 Mar 15;9(10):9184-94.
  3. Yu H, Wu M, Duan G, Gong X. One-step fabrication of eco-friendly superhydrophobic fabrics for high-efficiency oil/water separation and oil spill cleanup. Nanoscale. 2022;14(4):1296-309.

Here is a reference to a recent article in Nanomaterials.

Song J, Liu N, Li J, Cao Y, Cao H. Facile fabrication of highly hydrophobic onion-like candle soot-coated mesh for durable oil/water separation. Nanomaterials. 2022 Feb 24;12(5):761.

Response 3: The related references were cited in manuscript in line 42-43 as reference 11-15:

[11] Xue, C.H.; Ji, P.T.; Zhang, P.; Li, Y.R.; Jia, S.T. Fabrication of superhydrophobic and superoleophilic textiles for oil–water separation. Appl. Surf. Sci. 2013, 284, 464-471.

[12] Zhou, C.; Chen, Z.; Yang, H.; Hou, K.; Zeng, X.; Zheng, Y.; Cheng, J. Nature-inspired strategy toward superhydrophobic fabrics for versatile oil/water separation. ACS Appl. Mater. Interfaces 2017, 9, 9184-9194.

[13] Yu, H.; Wu, M.; Duan, G.; Gong, X. One-step fabrication of eco-friendly superhydrophobic fabrics for high-efficiency oil/water separation and oil spill cleanup. Nanoscale 2022, 14, 1296-1309.

[15] Song, J.; Liu, N.; Li, J.; Cao, Y.; Cao, H. Facile fabrication of highly hydrophobic onion-like candle soot-coated mesh for durable oil/water separation. Nanomaterials 2022, 12, 761.

Point 4: There are a number of problems with Figure 2.  Values for the WCA in the text(line 145) do not match with what is shown in Figure 2.  For example, the contact angle for the unmodified fabric is given as 0 degrees, but this is clearly not the case for Fig. 2g.   Three or four of the six starting materials in Figure 2 are clearly hydrophobic while the text indicate four of them are hydrophilic.  Figure 2e does not look like cotton. 

Response 4: Fig. 2 has been revised to match the WCA values in the text. The sequence of the substrates has been adjusted in Fig. 2.

Fig. 2 Photographs of original (left) and PPy/ZnO/HDTMS coated samples (right). a,b: corn stalk; c,d: sawdust; e,f: cotton; g,h: fabric; i,j: sponge; k,l: copper mesh.

Point 5: A native speaker of English should edit the manuscript.  There are issues with capitalization, punctuation, spacing, misspellings and awkward wording.

Response 5: The whole manuscript has been checked and revised.

Point 6: Define abbreviations given in the abstract.

Response 6: The abbreviations have been defined in abstract in line 17 and 19.

Point 7: In the Materials section, be more specific in your characterization of the substances to be modified.  Is the “cotton” sample raw fibers or a woven textile?   Specify the type of "fabric" used(e.g. polyester or silk) and give information on the fiber diameter and fiber density.   What is the chemical nature of the sponge?

Response 7: The cotton sample is the raw fiber. The fabric used is the cotton gauze, which has wrap and weft yarns of 32 counts of pure cotton with a number of 50×44 per square inch. The sponge sample used is melamine sponge. These substances have been introduced more specific in the materials section in line 89-92.

Point 8: Acetic acid does not appear in the materials list.  

Response 8: The acetic acid (CH3COOH,99.5%) was added in the materials list in line 94.

Point 9: XRD is mentioned, but no findings are reported.

Response 9: The XRD patterns were added in manuscript in line 198-206.

The XRD diffraction patterns of raw and coated fabric were shown in Fig. 4. The raw fabric had typical diffraction peaks of cellulose at 2θ= 14.9°, 16.5° and 22.9°, which also existed in coated fabric. The diffraction peaks of the coated fabric at 32.45°, 34.76°, 36.82° and 47.65° corresponded to the (100), (002), (101) and (102) lattice planes of ZnO nanoparticles, proving that the ZnO nanoparticles were successfully loaded on fabric surface. ZnO nanoparticles played a significant role in constituting the hierarchical structure for realizing the superhydrophobic properties.

Fig. 4 XRD patterns

Point 10: Line 38:  “…and application of low surface energy substances…”

Response 10: According to the suggestion, the sentence in line 38 has been revised as: “…and application of low surface energy substances…”

Point 11: Line 176:  Figure 4a presents reaction steps, not reaction mechanisms.

Response 11: The statement of Fig. 5 in line 217-218 has been revised as: Fig. 5 (a) Reaction steps of the PPy/ZnO/HDTMS coating; (b) A reaction mechanism diagram for HDTMS grafting on ZnO nanoparticles. The sentence in line 207 has been revised as: The reaction steps of the PPy/ZnO/HDTMS coating were shown in Fig. 5a.

Point 12: Line 178:  The high resolution SEM images might suggest that ZnO nucleates on the surface and grows in addition to precipitation or in lieu of precipitation.

Response 12: From high magnifications, ZnO nucleated on the surface and grew in addition to precipitation or in lieu of precipitation. The sentence was added in manuscript in line 151-152.

Reviewer 2 Report

The paper is devoted to a topical and very important topic - search for materials for oil-water separation. Obviously, the most promising are methods that combine simplicity and accessibility with environmental safety and a high level of product yield. On the other hand, it should be noted that both the nanocomposite polypyrrole – zinc oxide coatings and hexadecyltrimethoxysilane (HDTMS) are being actively studied at the moment.

 The authors have developed a technique for obtaining a new composite coating on six different surfaces.

Unfortunately, there is no information on the reproducibility of the technique and the adhesion of the coating to the substrate. Roughness is also an important characteristic in the study of surface wettability. These data are also important for the interpretation of the results. I recommend authors to pay attention to this in the future.

 It should be emphasized that the authors paid attention to studying the stability of the properties of the composite (fig. 7), which is almost always missing in the papers.

 From my point of view, in the Introduction it is necessary to add information on the study of HDTMS as highly hydrophobic material and its application in oil-water separation [1-3] and on other composites based on ZnO-HDTMS such as PAA-ZnO-HDTMS [4] and PU/PSF-ZnO-HDTMS [5].

 1. Pran Krisna Saha, Rony Mia, Yang Zhou & Taosif Ahmed. Functionalization of hydrophobic nonwoven cotton fabric for oil and water repellency. SN Applied Sciences 2021. V.3, Article number: 586

2. Feng Zhou , Minghui Yang , Yi Liu , Jiaxin Zhang , Yuting Gao , Chunjie Yan. Preparation of highly hydrophobic sepiolite for efficient oil removal. Microporous and Mesoporous Materials. 2022. V. 338, 111952

3. Jingming Zhao, Yuying Deng, Min Dai, Yanni Wu, Imran Ali, Changsheng Peng. Preparation of super-hydrophobic/super-oleophilic quartz sand filter for the application in oil-water separation. Journal of Water Process Engineering 46 (2022) 102561

4. Xiujuan Chen, Yunqiu Liu, Gordon Huang, Chunjiang An, Renfei Feng, Yao Yao, Wendy Huang & Shuqing Weng. Functional flax fiber with UV-induced switchable wettability for multipurpose oil-water separation. Frontiers of Environmental Science & Engineering. 2022. V.16. Article number: 153

5. Hongnan Zhang , Ke Li , Chenchen Yao , Jiatai Gu , Xiaohong Qin. Preparation of zinc oxide loaded polyurethane/polysulfone composite nanofiber membrane and study on its waterproof and moisture permeability properties. Colloids and Surfaces A: Physicochemical and Engineering Aspects 629 (2021) 127493

 There are several suggestions for the text of the Paper:

 Part 2.1 Materials  -you need to give full names and chemical formulas when describing materials (For example, Nanomaterials 2021, 11, 1688. https://doi.org/10.3390/nano11071688)

 Line 17 - Do not use an abbreviation in the abstract, give the full name of the compounds (PPy, HDTMS)

Line 65 - add the chemical formula of HDTMS

Line 79 - Indicate the purity of the HDTMS compound.

Line 161 – check the caption for figure 2 (g,h: fabric; i,j: sponge)

Line 164 – There are no XPD data in paper.

 Figure 3. FTIR spectrum

You write, that “the strong absorption peak at 3424 cm-1 were attributed to -OH stretching vibration.”.

For PPy: N-H group has stretching vibration in the area of 3500 – 3300 cm-1 and bending vibration at 1600 cm-1. And C-H group has stretching vibration in the area of 3100 cm-1.

 There is another interpretation of the spectra of PPy/ZnO in paper by Z. Huang, X. Li, C. Pan, P. Si, P. Huang, J. Zhou “Morphology-dependent electrochemical stability of electrodeposited polypyrrole/nano-ZnO composite coatings”, Materials Chemistry and Physics 279 (2022) 125775.

аnd

I.M Ibrahim, S Yunus, M.A Hashim.  “Relative Performance of Isopropylamine, Pyrrole and Pyridine as Corrosion Inhibitors for Carbon Steels in Saline Water at Mildly Elevated Temperatures”. International Journal of Scientific and Engineering Research, Volume 4, Issue 2, February-2013

 So, it is useful to add to Fig. 3 the IR spectra of the starting compounds (PPy, HDTMS)

Lines 167- 168 – add the references on IR spectra of PPy

Lines 172-173  - add the references on IR spectra of HDTMS

 In your case, the substrate (fabric) already has a complex IR spectrum. In the following works, add using a silicon or germanium substrate (as reference samples) that is transparent in the IR region.

For a correct interpretation, it is useful to make an IR spectrum at each layer of creating a composite. And to prove the presence of a layer of zinc oxide, add a study by method XRD.

 Please check for typos:

Line 78 - The beginning of a paragraph must start with a capital letter.

Fig. 3   misprint “fabic”

Author Response

Responses to Reviewer 2:

The paper is devoted to a topical and very important topic - search for materials for oil-water separation. Obviously, the most promising are methods that combine simplicity and accessibility with environmental safety and a high level of product yield. On the other hand, it should be noted that both the nanocomposite polypyrrole – zinc oxide coatings and hexadecyltrimethoxysilane (HDTMS) are being actively studied at the moment.

The authors have developed a technique for obtaining a new composite coating on six different surfaces.

Unfortunately, there is no information on the reproducibility of the technique and the adhesion of the coating to the substrate. Roughness is also an important characteristic in the study of surface wettability. These data are also important for the interpretation of the results. I recommend authors to pay attention to this in the future. It should be emphasized that the authors paid attention to studying the stability of the properties of the composite (fig. 7), which is almost always missing in the papers.

Point 1: From my point of view, in the Introduction it is necessary to add information on the study of HDTMS as highly hydrophobic material and its application in oil-water separation [1-3] and on other composites based on ZnO-HDTMS such as PAA-ZnO-HDTMS [4] and PU/PSF-ZnO-HDTMS [5].

  1. Pran Krisna Saha, Rony Mia, Yang Zhou & Taosif Ahmed. Functionalization of hydrophobic nonwoven cotton fabric for oil and water repellency. SN Applied Sciences 2021. V.3, Article number: 586
  2. Feng Zhou , Minghui Yang , Yi Liu , Jiaxin Zhang , Yuting Gao , Chunjie Yan. Preparation of highly hydrophobic sepiolite for efficient oil removal. Microporous and Mesoporous Materials. 2022. V. 338, 111952
  3. Jingming Zhao, Yuying Deng, Min Dai, Yanni Wu, Imran Ali, Changsheng Peng. Preparation of super-hydrophobic/super-oleophilic quartz sand filter for the application in oil-water separation. Journal of Water Process Engineering 46 (2022) 102561
  4. Xiujuan Chen, Yunqiu Liu, Gordon Huang, Chunjiang An, Renfei Feng, Yao Yao, Wendy Huang & Shuqing Weng. Functional flax fiber with UV-induced switchable wettability for multipurpose oil-water separation. Frontiers of Environmental Science & Engineering. 2022. V.16. Article number: 153
  5. Hongnan Zhang , Ke Li , Chenchen Yao , Jiatai Gu , Xiaohong Qin. Preparation of zinc oxide loaded polyurethane/polysulfone composite nanofiber membrane and study on its waterproof and moisture permeability properties. Colloids and Surfaces A: Physicochemical and Engineering Aspects 629 (2021) 127493

Response 1: In the Introduction, the study of HDTMS as highly hydrophobic material was added in line 65-71:

For realizing the low surface energy on rough surface, hexadecyltrimethoxysilane (HDTMS) is highly used for preparation of superhydrophobic material in oil-water separation [26-28]. Chen et al. [29] coated the flax fiber with ZnO-HDTMS using a plasma-grafted poly (acrylic acid) as the binding agent for efficient oil-water separation. Zhang et al. [30] bonded ZnO nanoparticles to the surface of polyurethane/polysulfone with hydrophobically modification of HDTMS to prepare a new waterproof and moisture permeable membrane.

  1. Saha, P.K.; Mia, R.; Zhou Y.; Ahmed, T. Functionalization of hydrophobic nonwoven cotton fabric for oil and water repellency. SN Appl. Sci. 2021, 3, 586.
  2. Zhou, F.; Yang, M.; Liu Y.; Zhang, J.; Gao, Y.; Yan, C. Preparation of highly hydrophobic sepiolite for efficient oil removal. Micropor. Mesopor. Mat. 2022, 338, 111952.
  3. Zhao, J.; Deng, Y.; Dai, M.; Wu, Y.; Ali, I.; Peng, C. Preparation of super-hydrophobic/super-oleophilic quartz sand filter for the application in oil-water separation. J. Water Process Eng. 2022, 46, 102561.
  4. Chen, X.; Liu, Y.; Huang, G.; An, C.; Feng, R.; Yao, Y.; Huang, W.; Weng, S. Functional flax fiber with UV-induced switchable wettability for multipurpose oil-water separation. Front. Env. Sci. Eng. 2022,16, 153.
  5. Zhang, H.; Li, K.; Yao, C.; Gu, J.; Qin, X. Preparation of zinc oxide loaded polyurethane/polysulfone composite nanofiber membrane and study on its waterproof and moisture permeability properties. Colloid Surfaces A 2021, 629, 127493.

Point 2: There are several suggestions for the text of the Paper:

 Part 2.1 Materials  -you need to give full names and chemical formulas when describing materials (For example, Nanomaterials 2021, 11, 1688. https://doi.org/10.3390/nano11071688).

Response 2: In Part 2.1, the full names and chemical formulas have been given when describing materials. This part has been revised in line 92-99 in manuscript as follows:

Zinc chloride (ZnCl2, 99%), pyrrole (C4H4NH, 99%), hexadecyltrimethoxysilane (HDTMS, ≥85%), ammonium hydroxide (NH3∙H2O, 28%), absolute ethyl alcohol (C2H5OH), acetic acid (CH3COOH, 99.5%), ethyl acetate (C4H8O2, 99%), liquid paraffin (99%), benzene (C6H6, ≥99.5%), petroleum ether, tetrachloromethane (CCl4, 98%) methylbenzene (C7H8, 99%), chloroform (CHCl3, 99%) and dichloromethane (CH2Cl2, 99%) were acquired from Shanghai Macklin Biochemical Co., Ltd. Sodium hydroxide (NaOH, ≥ 96.0%), hydrochloric acid (HCl, 36.5% ∼38%), ferric trichloride (FeCl3, 99%) were purchased from Tianjin Kermel Chemical Reagent Co., Ltd.

Point 3: Line 17 - Do not use an abbreviation in the abstract, give the full name of the compounds (PPy, HDTMS)

Response 3: The full names of PPy and HDTMS have been given in abstract in line 17.

Point 4: Line 65 - add the chemical formula of HDTMS

Response 4: The chemical formular of HDTMS was added in manuscript in line 93, which was expressed as CH3(CH2)15Si(OCH3)3.

Point 5: Line 79 - Indicate the purity of the HDTMS compound.

Response 5: The purity of HDTMS is ≥85%, and it is indicated in manuscript in line 93.

Point 6: Line 161 – check the caption for figure 2 (g,h: fabric; i,j: sponge).

Response 6: The sequence of the substrate in Fig. 2 has been adjusted in accordance with the caption. The Fig. 2 has been revised as follows:

Fig. 2 Photographs of original (left) and PPy/ZnO/HDTMS coated samples (right). a,b: corn stalk; c,d: sawdust; e,f: cotton; g,h: fabric; i,j: sponge; k,l: copper mesh.

Point 7: Line 164 – There are no XRD data in paper.

Response 7: The XRD patterns were added in manuscript in line 198-206.

Point 8: Figure 3. FTIR spectrum

You write, that “the strong absorption peak at 3424 cm-1 were attributed to -OH stretching vibration.”.

For PPy: N-H group has stretching vibration in the area of 3500 – 3300 cm-1 and bending vibration at 1600 cm-1. And C-H group has stretching vibration in the area of 3100 cm-1.

 There is another interpretation of the spectra of PPy/ZnO in paper by Z. Huang, X. Li, C. Pan, P. Si, P. Huang, J. Zhou “Morphology-dependent electrochemical stability of electrodeposited polypyrrole/nano-ZnO composite coatings”, Materials Chemistry and Physics 279 (2022) 125775.

аnd

I.M Ibrahim, S Yunus, M.A Hashim.  “Relative Performance of Isopropylamine, Pyrrole and Pyridine as Corrosion Inhibitors for Carbon Steels in Saline Water at Mildly Elevated Temperatures”. International Journal of Scientific and Engineering Research, Volume 4, Issue 2, February-2013

 So, it is useful to add to Fig. 3 the IR spectra of the starting compounds (PPy, HDTMS)

Lines 167- 168 – add the references on IR spectra of PPy

Lines 172-173  - add the references on IR spectra of HDTMS

 In your case, the substrate (fabric) already has a complex IR spectrum. In the following works, add using a silicon or germanium substrate (as reference samples) that is transparent in the IR region.

Response 8: According to the suggestion, the FTIR analysis in line 183-197 has been revised as follows:

As shown in Fig. 3, the FTIR spectra were tested in order to ensure the chemical compositions of the prepared coating. For raw fabric, the strong absorption peak at 3424 cm-1 were attributed to -OH stretching vibration. The -OH stretching vibration was greatly reduced after coating treatment due to the modification of HDTMS. The FTIR spectrum for PPy [37-39] and HDTMS [40] was shown in Table 2. The characteristic peaks at 1685, 1558 and 1315 cm-1 in coated fabric were assigned to the C=N, C=C, C–N in PPy. Two absorption peaks at 2851 cm-1 and 2921 cm-1 stemmed from C–H stretching vibration of –CH2– and –CH3. The adsorption at 2918 cm-1 (C–H stretching vibration) in original fabric split into two peaks (2921 and 2851 cm-1) after coating, which was attributed to HDTMS modification. Furthermore, the stretching vibration of Si–C was observed at 781 cm-1, demonstrating that the HDTMS was successfully grafted onto the PPy/ZnO coated fabric.  

  1. Huang, Z.; Li, X.; Pan, C.; Si, P.; Huang, P.; Zhou, J. Morphology-dependent electrochemical stability of electrodeposited polypyrrole/nano-ZnO composite coatings. Mater. Chem. Phys. 2022, 279, 125775.
  2. Ibrahim, I.M.; Yunus, S.; Hashim, M.A. Relative performance of isopropylamine, pyrrole and pyridine as corrosion inhibitors for carbon steels in saline water at mildly elevated temperatures. Int. J. Eng. Sci. 2013, 4, 1-12.
  3. Chougule, M.A.; Pawar, S.G.; Godse, P.R.; Mulik, R.N.; Sen, S.; Patil, V.B. Synthesis and characterization of polypyrrole (PPy) thin films. Soft Nanosci. Let. 2011, 1, 6-10.
  4. Liu, L.; Chen, Y.; Huang, P. Preparation and tribological properties of organically modified graphite oxide in liquid paraffin at ultra-low concentrations. RSC Adv. 2015, 5, 90525.

Table 2. The FTIR spectrum for PPy and HDTMS.

Wavelength (cm-1)

Attribution of the characteristic peaks

PPy

3522

N – H bonding in the molecule

1685

C=N bonds in the molecule

1558

1437

C=C bonds in the molecule

1315

C-N bonds in the molecule

811

920

Bonding of C – H in the molecule

HDTMS

2921

Asymmetric stretching vibration of –CH2

2851

Symmetric stretching vibration of –CH2

1000-1100

Bending vibration of Si–O–C

781

Stretching vibration of Si–C

According to the suggestion, we will use a substrate that is transparent in the IR region to analysis the coating in the future.

Point 9: For a correct interpretation, it is useful to make an IR spectrum at each layer of creating a composite. And to prove the presence of a layer of zinc oxide, add a study by method XRD.

Response 9: The FTIR spectrum for PPy and HDTMS was shown in Table 2 for interpretation of the superhydrophobic coating. The XRD patterns were added in manuscript in line 198-206:

The XRD diffraction patterns of raw and coated fabric were shown in Fig. 4. The raw fabric had typical diffraction peaks of cellulose at 2θ= 14.9°, 16.5° and 22.9°, which also existed in coated fabric. The diffraction peaks of the coated fabric at 32.45°, 34.76°, 36.82°and 47.65°corresponded to the (100), (002), (101) and (102) lattice planes of ZnO nanoparticles, proving that the ZnO nanoparticles were successfully loaded on fabric surface. ZnO nanoparticles played a significant role in constituting the hierarchical structure for realizing the superhydrophobic properties.

Fig. 4 XRD patterns

Point 10: Line 78 - The beginning of a paragraph must start with a capital letter.

Response 10: The “corn stalk” has been revised as “Corn stalk” in line 89 in manuscript.

Point 11: Fig. 3   misprint “fabic”

Response 11: The misprint in Fig. 3 has been corrected as follows:

 Fig. 3 FTIR spectrum

Reviewer 3 Report

-          At Introduction, specify the practical applications (line 60).

-          At 2.1., line 79, mention the full name for HDTMS.

-          At 2.4 Characterization, mention more information about the characterization of materials, like:

For FTIR: Data were collected at room temperature, in the 4000−400 cm−1 range (30 scans at a resolution of 4 cm−1)

For WTA: The water drop volume selected to measure the contact angles was of 6 µL.

-          At Table 1, include the standard deviation (e.g.: 152.7 ± 5)

-          In Figure 3, include the important peaks.

-          At Conclusions, specify the practical applications.

Author Response

Responses to reviewer 3:    

Point 1: At Introduction, specify the practical applications (line 60).

Response 1: Increasing water pollution from offshore oil spills and industry activities has threatened the human beings and environment seriously. The superhydrophobic material has attracted significant attention in oil/water separation. Thus, the practical applications have been specified in oil/water separation from chemical leakages and oil spills in line 63-64 in manuscript.

Point 2: At 2.1., line 79, mention the full name for HDTMS.

Response 2: “Hexadecyltrimethoxysilane”, the full name for HDTMS, was added in line 92-93 in manuscript.

Point 3: At 2.4 Characterization, mention more information about the characterization of materials, like:

For FTIR: Data were collected at room temperature, in the 4000−400 cm−1 range (30 scans at a resolution of 4 cm−1).

For WTA: The water drop volume selected to measure the contact angles was of 6 µL.

Response 3: At 2.4 Characterization, more information about the characterization of materials has been mentioned in line 137-139 and 142-143. For WTA: The WCA was measured using a 5 μL of deionized water droplet, which was determined by averaging 5 values from different positions of each sample. For FTIR: The FTIR data were collected at room temperature, in the 4000-400 cm−1 range (30 scans at a resolution of 4 cm−1).

Point 4: At Table 1, include the standard deviation (e.g.: 152.7 ± 5)

Response 4: At Table 1, the standard deviation has been added in line 171. Table 1 has been revised as:

Table 1. Calculated data based on the Cassie-Baxter equation.

Substrate

WCA

SA

Air fraction(1-fs)

corn stalk/PPy/ZnO/HDTMS

152.7±1.6

2.0±0.6

0.88

sawdust/PPy/ZnO/HDTMS

153.9±2.3

3.5±1.5

0.89

cotton/PPy/ZnO/HDTMS

153.3±1.1

3.0±1.7

0.88

fabric/PPy/ZnO/HDTMS

157.6±4.5

3.8±1.8

0.92

sponge/PPy/ZnO/HDTMS

163.1±4.6

2.3±0.8

0.95

copper mesh/PPy/ZnO/HDTMS

162.1±3.3

3.9±2.3

0.95

Point 5: In Figure 3, include the important peaks.

Response 5: The important peaks have been added in Fig. 3. Fig. 3 has been revised as follows:

Fig. 3 FTIR spectrum

Point 6: At Conclusions, specify the practical applications.

Response 6: The application of the superhydrophobic material has been specified in conclusion in line 345-349 as follows: The superhydrophobic material prepared in this work displayed outstanding performance for removing/ collecting oil contaminants. This strategy for achieving superhydrophobic material is simple, feasible and highly efficient, offering a wide range of potential applications in reclaiming the oil/water mixtures from chemical leakages and oil spills.

Round 2

Reviewer 1 Report

The authors have a done a good job responding to major points with the exception of the "separation efficiency".  Their measure as defined by Equation 1 is not an acceptable measure of separation efficiency.  As noted previously, it is a misleading gauge of performance.  This is an important point given the emphasis on oil/water separation in this work. As such, I am indicating this as a major revision.  Otherwise the authors have addressed points of concern.

The authors can redefine the quantity "eta" in equation 1 to something like "oil filtrate fraction", but not cannot use separation efficiency in the text and Figures 7 and 8.  The alternative would be to run additional tests over a greater range of compositions and measure water concentrations for an expression addressing the problems of their definition of "separation efficience".  

Author Response

Point 1: The authors have a done a good job responding to major points with the exception of the "separation efficiency".  Their measure as defined by Equation 1 is not an acceptable measure of separation efficiency.  As noted previously, it is a misleading gauge of performance.  This is an important point given the emphasis on oil/water separation in this work. As such, I am indicating this as a major revision. Otherwise the authors have addressed points of concern.

The authors can redefine the quantity "eta" in equation 1 to something like "oil filtrate fraction", but not cannot use separation efficiency in the text and Figures 7 and 8.  The alternative would be to run additional tests over a greater range of compositions and measure water concentrations for an expression addressing the problems of their definition of "separation efficiency".  

Response 1: According to the suggestion, the quantity "η" in equation 1 is redefined as "oil filtrate fraction". The "oil filtrate fraction" has been used to replace "separation efficiency" in the text and Figures 7 and 8 in manuscript.

Reviewer 3 Report

Dear Sirs,

The manuscript was improved and it can be publish in this form.

Author Response

The reviewer has agreed to the publication in this form. Thank you very much.

Round 3

Reviewer 1 Report

With the change in definition for "eta", the manuscript is now acceptable for publication in Nanomaterials.